# Comprehensive Serum Glycopeptide Spectra Analysis Combined with Artificial Intelligence (CSGSA-AI) to Diagnose Early-Stage Ovarian Cancer

**DOI:** 10.3390/cancers12092373

**Published:** 2020-08-21

**Authors:** Kazuhiro Tanabe, Masae Ikeda, Masaru Hayashi, Koji Matsuo, Miwa Yasaka, Hiroko Machida, Masako Shida, Tomoko Katahira, Tadashi Imanishi, Takeshi Hirasawa, Kenji Sato, Hiroshi Yoshida, Mikio Mikami

**Affiliations:** 1Medical Solution Promotion Department, Medical Solution Segment, LSI Medience Corporation, Tokyo 1748555, Japan; katahira.tomoko@ma.medience.co.jp; 2Research Supporting Department, Kyushu Pro Search Limited Liability Partnership, Fukuoka 8190388, Japan; 3Department of Obstetrics and Gynecology, Tokai University School of Medicine, Kanagawa 2591193, Japan; ikedam@tokai-u.jp (M.I.); masaru.hayashi@tokai.ac.jp (M.H.); yasakamiwa@tokai.ac.jp (M.Y.); hiroko.machida@tokai.ac.jp (H.M.); shida@is.icc.u-tokai.ac.jp (M.S.); hira@is.icc.u-tokai.ac.jp (T.H.); igymichael@gmail.com (K.S.); hiroshiyoshidamd@gmail.com (H.Y.); 4Division of Gynecologic Oncology, Department of Obstetrics and Gynecology, University of Southern California, Los Angeles, CA 90033, USA; koji.matsuo@med.usc.edu; 5Norris Comprehensive Cancer Center, University of Southern California, Los Angeles, CA 90033, USA; 6Department of Molecular Life Science, Division of Basic Medical Science and Molecular Medicine, Tokai University School of Medicine, Kanagawa 2591193, Japan; imanishi@tokai-u.jp

**Keywords:** AI, deep learning, ovarian cancer, screening, early stage, convolutional neural network, glycopeptide, mass spectrometry, glycomics, AlexNet

## Abstract

Ovarian cancer is a leading cause of deaths among gynecological cancers, and a method to detect early-stage epithelial ovarian cancer (EOC) is urgently needed. We aimed to develop an artificial intelligence (AI)-based comprehensive serum glycopeptide spectra analysis (CSGSA-AI) method in combination with convolutional neural network (CNN) to detect aberrant glycans in serum samples of patients with EOC. We converted serum glycopeptide expression patterns into two-dimensional (2D) barcodes to let CNN learn and distinguish between EOC and non-EOC. CNN was trained using 60% samples and validated using 40% samples. We observed that principal component analysis-based alignment of glycopeptides to generate 2D barcodes significantly increased the diagnostic accuracy (88%) of the method. When CNN was trained with 2D barcodes colored on the basis of serum levels of CA125 and HE4, a diagnostic accuracy of 95% was achieved. We believe that this simple and low-cost method will increase the detection of EOC.

## 1. Introduction

Ovarian cancer is one of the most common malignancies worldwide. Although the 5-year average annual death rate due to ovarian cancer decreased by 2.3% during 2011–2015 [1], epithelial ovarian cancer (EOC) remains the leading cause of death among gynecological cancers [2,3]. According to the National Cancer Institute [4], among Caucasians, the 5-year survival rate has improved from 37% to 45% from 1975 to 2006, whereas that among African-Americans has worsened from 43% to 37% during the same period. Diagnostics for early-stage EOC are urgently needed because more than 50% of symptomatic women are diagnosed at an advanced stage and the 5-year survival rate of patients with advanced stage EOC is less than 30% [5]. Imaging technologies, such as positron emission tomography/computed tomography (PET/CT) with fluorodeoxyglucose, are useful in detecting early-stage EOC that is less than 3 cm in size; however, it is not widely employed for screening because of examinee’s economical and physical burdens [6,7]. On the contrary, a convenient blood test using tumor markers, such as cancer antigen 125 (CA125), is partially used for screening of EOC [8,9]. However, the sensitivity and specificity of the CA125 test are not sufficient to detect early-stage cancer, and it does not alleviate the mortality rate [10,11]. Recent advances in the detection of early-stage EOC, such as micro RNA or cell-free DNA detection, have shown significant improvement in sensitivity and specificity [12,13]. However, these technologies are only partially successful in detecting early-stage EOC, because they cannot distinguish EOC from benign ovarian tumor, which has a much higher prevalence than that of EOC. To establish a reliable and robust blood test for early-stage EOC detection, we developed a glycopeptide profiling technology using mass spectrometry, which detects aberrant serum sugar chains in patient sera [14]. In our previous studies, we have shown that C4-binding protein with fully sialylated sugar chains is specifically increased in the sera of EOC patients [15,16,17]. Based on this, we developed a comprehensive serum glycopeptide spectra analysis (CSGSA) technology by combining glycopeptide profiling with orthogonal partial least squares discriminant analysis (OPLS-DA) [18]. This approach has remarkably improved the diagnostic potential of this test to distinguish between EOC and non-EOC.

Recently, artificial intelligence (AI), in particular image recognition using convolutional neural network (CNN), has emerged to distinguish objects in images [19], and now contributes on pedestrian recognition [20], surveillance videos [21] or financial forecasting [22]. CNN has multiple process layers (steps) and optimizes parameters assigned in each layer to achieve correct recognition of target objects using a lot of training samples [23,24]. CNN can not only classify images more accurately than conventional neural networks or machine-learning systems [25] but also transfer the acquired knowledges to other studies to reduce training effort [26]. The “transfer learning” technology has enabled clinical scientists to utilize AI in studies with a limited number of cases [25,27]. Oh et al. [25] used CNN to learn 695 brain images of Alzheimer’s disease patients acquired by magnetic resonance imaging (MRI) to distinguish it from normal brain and achieved 87% accuracy. Since preparing a large number of clinical specimens is difficult, this transfer learning technology is meaningful to accelerate clinical studies using AI [28]. Moreover, since the development of CNN is outstanding among general deep learning technologies, some scientists have attempted to convert numerical (non-image) data into images and let CNN learn the images. Sharma et al. [29] have developed a tool named DeepInsight, which converts non-image numerical data, such as RNA-seq, into well-organized images that allow CNN to learn and distinguish phenotypes.

AlexNet is a pretrained CNN architecture developed by Krizhevsky [30], who won the first prize in the ImageNet Large Scale Visual Recognition Challenge (LSRVC) contest in 2012. AlexNet has 25 layers, which include five convolutional layers interleaved with max pooling layers and local response normalization layers. Each layer consists of sets of feature maps, which are the responses of a filter on the output of the preceding layer. In this study, we aimed to use AlexNet, the latest CNN-based technology, as a discrimination tool for CSGSA to identify early-stage EOC. To facilitate CNN training, we converted numerical data of glycopeptide expression to 2D barcode images and let CNN learn and distinguish early-stage EOC.

## 2. Results

### 2.1. The Training Options

The training options, namely initial learning rate (speed of convergence), maximum number of epochs (a full pass of the data), and minimum batch size (a size of data subunit), were chosen in such a way that the calculation is fast and accurate. When the initial learning rate is too low or too high, training takes a long time or does not reach the correct solution. If the number of epochs is too small, calculation stops before converging, and if the minimum batch size is too big (i.e., number of batches is too small), calculating cost through the network becomes expensive, i.e., the efficacy of optimization becomes worse. When the initial learning rate was set at 0.0001, maximum number of epochs at 30, and minimum batch size at 5, the training converged within acceptable period (10 to 20 min) without diverging, and the accuracy and loss curves reached a plateau before calculation stopped, indicating that the maximum number of epochs was adequate (Appendix A). Thus, we performed the following experiments with these values.

Two optimizers—adaptive moment estimation (adam) and stochastic gradient descent with momentum (sgdm)—were evaluated regarding how they correctly distinguish EOC from non-EOC. Each model was trained using 60% of data and evaluated using the remaining 40% of data. This process was repeated 10 times by randomly reselecting training data sets, and ROC-AUCs (area under curve of receiver operating characteristic) were obtained from predicted values of test samples. As a result, there was no significant difference between the two optimizers (adam: 0.878, sgdm: 0.881). Therefore, we used sgdm for the following experiments.

### 2.2. Learning Efficacy of Pretrained CNN Models

The diagnostic performance of two AlexNet models, non-pretrained and pretrained (transferred), was compared. The purpose of this evaluation was to clarify the effectiveness of pretraining, even if it was not related to actual training (determination of EOC). The non-pretrained AlexNet model was established using AlexNet framework involving fresh 25 layers without any training. The number of weighs, bias, slide, and padding were reproduced according to an original AlexNet framework (Table 1 [30]). The transferred AlexNet model was based on a pretrained AlexNet model that was already trained by more than one million images from the ImageNet database [31]. This model was able to categorize 1000 objects, such as keyboard, mouse, pencil, and many animals. Only the last three layers, namely fully connected layer, softmax layer, and classification layer, were replaced with new fresh layers.

The layers used in the two models are illustrated in Figure 1a. The gray layers indicate that they were already pretrained and acquired image features, and the parameters are not changed during additional training. On the contrary, the white layers indicate that their parameters were not pretrained and were optimized during training. To evaluate the diagnostic performance of the two models, we calculated ROC-AUCs using the repeated holdout validation method as described next. First, the two models were trained by 60% of the 351 samples and evaluated by the remaining 40% samples, and this process was repeated 10 times by randomly reselecting 60% training samples. ROC-AUCs were calculated by predicted values obtained from the test samples. Wilcoxon rank sum test was deployed to assess the statistical difference in AUC values as the two groups were independent. As a result, the average ROC-AUC of the transferred AlexNet model reached 0.881, whereas that of the non-pretrained AlexNet model was 0.853 (Figure 1b). Further, the *p*-value of the Wilcoxon rank sum test between the two models was 0.020 (Figure 1c). This suggests that even an irrelevant experience, for example, recognizing keyboard, mouse, and pencil, increases learning efficacy for recognizing 2D barcodes of EOC glycopeptide patterns. Additionally, it indicates that the latent features of images are partially common among real objects and artificially generated 2D barcodes.

### 2.3. Glycopeptide Alignment to Generate 2D Barcodes

To understand how glycopeptide alignment to generate a 2D barcode affects diagnostic performance, we prepared two sets of 2D barcodes and compared their diagnostic performances. One set was aligned by the order of liquid chromatography elution time (Rt-based barcode), and the other was aligned by the order of principal component analysis (PCA) loading values (PCA-based barcode, Figure 2a). Considering that CNN recognizes spatial relative distances among object parts, for example, CNN recognizes a human face by identifying relative distances between mouth, nose, and eyes, CNN would recognize EOC features more precisely if the glycopeptides with similar expression pattern are closely located. Since PCA classifies glycopeptides based on their expression similarity, we used PCA to determine the positions of glycopeptides in a 2D barcode. PCA-based barcode was generated as follows: first, PCA of 1712 glycopeptides from 351 samples was performed, and the loading values of the first and second components for 1712 glycopeptides were obtained. Then, 1712 glycopeptides were sorted based on the first loading values, followed by dividing them into 41 groups. Further, they were sorted based on the second loading values within each group. The X coordinate in a 2D barcode indicated the number of groups and the Y coordinate indicated the order in each group. We used Rt-based barcode as a control for PCA-based barcode, assuming that Rt-based barcode is estimated as random glycopeptide alignment, because there is no relation between elution order and expression similarity. Rt-based barcode was generated by aligning glycopeptides from left-top to right-down based on the order of elution time. The method of how to change black and white contrast is shown in Appendix A. Since a 41 × 42 matrix has 1722 cells, and 10 blanks are generated on the bottom right corner with the expression values of 1712 glycopeptides, the blank cells were replaced with white. The typical 2D barcodes of EOC and non-EOC generated by Rt-based alignment and PCA-based alignment are shown in Figure 2b. It can be observed that black dots (strong expression) are more converged in the PCA-based barcode than in the Rt-based barcode. Training and testing were performed using the transferred AlexNet model, and the diagnostic performance was evaluated with the repeated holdout validation method. As a result, CNN was able to recognize early-stage EOC more precisely using PCA-based barcodes (AUC: 0.881) than using Rt-based barcodes (AUC: 0.852) (Figure 2c). The *p*-value of the Wilcoxon rank sum test between the two models was 0.024 (Figure 2d). Overall, this exercise indicated that CNN could elicit EOC features more precisely when it is trained by PCA-based barcodes.

### 2.4. Training with 2D Barcodes Colored by CA125 and HE4

To further enhance the learning efficacy and diagnostic performance of CNN, we added CA125 and HE4 information into the 2D barcode by changing the color (multicolored model). When the level of CA125 in the serum increases, the red level of the barcode decreases, and when the serum level of HE4 increases, the green level of the barcode decreases (Figure 3a). Appendix A shows how to color the barcode. Ten blanks generated on the bottom right corner were filled with the colors defined by both CA125 and HE4. The barcodes before and after coloring on the basis of CA125 and HE4 levels are shown in Figure 3b,c. It can be observed that the 2D barcodes of EOC tend to be blue and those of non-EOC tend to light purple depending on CA125 and HE4 levels. Training and testing were performed using the transferred and PCA-based AlexNet model and the diagnostic performance was evaluated with the repeated holdout validation method. As a result, CNN classified the multicolored 2D barcodes by AUC of 0.954 (Figure 3d), and the *p*-value of the Wilcoxon rank sum test was less than 10^-3^ between multicolored and non-multicolored models (without CA125 and HE4 information, Figure 3e). Since ROC-AUCs of CA125 and HE4 were 0.86 and 0.87, respectively, the combination successfully increased the diagnostic performance of CNN.

### 2.5. CSGSA-AI Scheme and its Diagnostic Performance

The scheme of CSGSA combined with AI (CSGSA-AI) is shown in Figure 4. Serum glycoproteins obtained from examinees were digested into peptides or glycopeptides using a protease (trypsin).

After glycopeptides were enriched by ultrafiltration, they were analyzed using liquid chromatography quadruple time-of-flight mass spectrometry (LC-QTOF-MS) to obtain the glycopeptide expression pattern. This pattern was converted into a 2D barcode (Appendix A). When the 2D barcode was generated, glycopeptides were aligned by the order of PCA loadings, and the color was determined based on the levels of CA125 and HE4. The CNN model, which was previously trained by supervised samples and acquired EOC discrimination ability, could now determine unknown 2D barcodes generated from asymptomatic participants and distinguished between EOC or non-EOC. It took two days to complete diagnosis, and 50 samples could be analyzed within one day with one mass spectrometer. The diagnostic performance of CSGSA-AI is shown in Table 2. The sensitivity, specificity, positive predictive value, and negative predictive value were 79%, 96%, 89%, and 92%, respectively. Considering that the EOC group consisted of only stage I patients, this performance is acceptable for the screening of early-stage EOC.

## 3. Discussion

In our previous study, we developed a tool that combined CSGSA with OPLS-DA to provide accurate recognition of early-stage EOC. OPLS-DA is based on a liner algebraic, which features robust and less over-fitting; however, distinguishing complicated cases remains challenging due to its limited separation ability. On the contrary, deep learning powered by CNN is based on a non-linear approach that includes several parameters; therefore, it is advantageous for eliciting features to distinguish specific categories. The biggest challenge to introduce CNN into EOC recognition is to collect a large number of samples to let CNN learn. The number of samples is generally acknowledged as 1000 and more. It is not an easy task to accomplish, because the number of EOC cases is limited compared to those of other cancers. To circumvent this problem, we used a pretrained CNN model, which was already trained with more than 1 million images of 1000 objects, such as strawberry, blueberry, orange, lemon, cosmos, or dahlia, provided by the ImageNet LSVRC contest [31]. Since the underlying features of these objects are partially common, they can be utilized for classifying other kinds of images, such as mass spectrometry data patterns. In this study, we showed that the pretrained CNN model AlexNet can recognize early-stage EOC using PCA-based 2D barcodes at the ROC of 0.881. The ROC further increased to 0.951 when 2D barcodes were generated based on the serum levels of CA125 and HE4. We emphasize that the EOC samples used in this study were all from patients with early-stage (stage I) cancer, whose 5-year survival rates is over 90% [5]. As this image-based diagnostic technique puts much less economical and physical burden on the patients than the classical blood test, this method will contribute to increase in the number of medical checks and decrease in the number of patients who develop advanced stage cancer. Nonetheless, to progress this study further, we have to overcome some issues. First, the AUCs of ROC of training set were almost 100%, indicating that the training has mostly caused overfitting; in other words, the amount of supervised data was not sufficient. Therefore, CNN does not completely elicit the underlying features. Second, the AUC of ROC (0.951) was acceptable but not enough to introduce this technology into actual screening of EOC. Since the morbidity of EOC is 10 to 20 per 100,000 population, the positive predicted value (PPV) becomes less than 1% if the sensitivity of the diagnostics is more than 50%. To improve the PPV, other architectures, such as Resnet50 (https://mathworks.com/help/deeplearning/ref/resnet50.html), VGG-16 (https://mathworks.com/help/deeplearning/ref/vgg16.html), VGG-19 (https://mathworks.com/help/deeplearning/ref/vgg19.html), and Googlenet (https://mathworks.com/help/deeplearning/ref/googlenet.html), can be used instead of AlexNet. These architectures can elicit specific features more precisely than AlexNet; however, the number of parameters that should be optimized is much higher for these architectures than that for AlexNet. Therefore, more samples are required to use these architectures, even if pretrained (transferred) models are used.

Deep learning technology is further evolving, and many new technologies have recently emerged. One of the most promising technology is generative adversarial networks [32], which aims to generate new images that have never existed, but everyone can recognize them as real. Generative adversarial networks consist of two functions: “generator” and “discriminator”. Generator randomly generates new images while evolving itself (optimizing its parameters) to create new images that discriminator misclassifies as real. Contrarily, discriminator also evolves itself by optimizing its parameters to distinguish real from artificial images that generator creates. When this process is repeated many times, the generator finally creates new images that everyone recognizes to be real. The fundamental difficulty in clinical studies is to obtain sufficient number of samples; therefore, this technology will overcome the problem to create the required number of samples and lead to more robust and more reproducible analysis. Marouf et al. [33] proposed the conditional single-cell generative adversarial neural network (cscGAN) for the realistic generation of single-cell RNA-seq data, and showed that cscGAN outperforms existing methods for single-cell RNA-seq data generation in quality and holds great promise for the augmentation of other biomedical data types. When this technology is applied to CSGSA-AI, Generative adversarial networks will create new 2D barcodes of EOC or non-EOC, which will supplement the limited number of cases, resulting in increasing the accuracy of diagnosis. Moreover, if the generator plays the role of a “separator” of unsupervised 2D-barcodes obtained from examinees, CSGSA-AI will increase their accuracy of diagnosis while the tests are being conducted. This indicates that the more the CSGSA-AI conducts tests, the more the diagnostic accuracy increases, which is just what we expect from AI.

Some issues we should address remain before we realize this technology as a clinical test. It is known that the levels of some glycoproteins in serum are influenced by diet, sex, or circadian activities; therefore, the stability of the target glycopeptides should be validated.

## 4. Materials and Methods

### 4.1. Patient Samples

A total of 97 serum samples were collected from patients with early-stage EOC at the time of ovarian mass detection prior to the initiation of any treatment (stage I). The EOC diagnosis and stage classification were performed by FIGO guidelines [34]. The average age of these patients was 54.4 (SD ± 12.9) years. The non-EOC control group (*n* = 254), whose average age was 53.8 (SD ± 11.4) years, comprised both healthy women (*n* = 220) and women with gynecologic diseases other than ovarian cancer (*n* = 34). Among the 34 patients with gynecologic disease, 20 had uterine fibroid and 14 had ovarian cysts (Table 3). The sera of patients with EOC and those with other gynecologic diseases were obtained from Tokai University Hospital (Kanagawa, Japan). Those of healthy women were obtained from SOIKEN (Osaka, Japan), KAC (Kyoto, Japan), and Sanfco (Tokyo, Japan) with informed consent from each patient or volunteer. Study specific-exclusion criteria are shown in Appendix A. All blood samples were collected by venous puncture before surgery or any treatment. The sera were separated from the blood by centrifugation and stored at −80 °C until further examination. Quality control (QC) sample was prepared with pooling approximately 10 EOC and 10 non-EOC sera (total 5 mL), which was dispensed into microtubes 20 µL each and stored at −80 °C. Two QC samples were analyzed with 20 patient samples to standardize the glycopeptide expression levels of each patient.

### 4.2. Study Approval

Institutional Review Board (IRB) of Tokai University approved the use of patient clinical information and serum/tumor samples (IRB registration number: 09R-082, approved on 17 September 2009).

### 4.3. Sample Preparation

Ten microliters of fetal calf fetuin (Sigma, St. Louis, MO, USA; 2 mg/mL) aqueous solution was added to the serum sample (20 μL) to check the efficiency of trypsin digestion or the recovery of glycopeptides. Then, acetone containing 10% of trichloroacetic acid (120 μL, Wako Pure Chemical Industries, Ltd., Osaka, Japan) was added to remove serum albumin. After mixing and centrifuging at 12,000 rpm for 5 min, the supernatant was removed and cooled acetone (400 μL) was added to wash the precipitate. The mixture was centrifuged at 10,000× *g* for 5 min and the supernatant was removed again. The precipitate was mixed with a denaturing solution that was prepared by mixing urea (80 μg, Wako Pure Chemical Industries, Ltd., Osaka, Japan), Tris-HCl buffer (pH 8.5, 100 μL), 0.1 M EDTA (10 μL), 1 M Tris (2-carboxyethyl) phosphine hydrochloride (5 μL, Sigma, St. Louis, MO, USA), and water (38 μL), and the proteins were denatured for 10 min at 37 °C. Subsequently, 1 M 2-iodoacetamide (40 μL, Wako Pure Chemical Industries, Ltd., Osaka, Japan) was added to the denaturing solution to protect thiol residues in proteins. The mixture was incubated for 10 min at 37 °C in the dark. The solution was transferred into a 30 kDa ultrafiltration tube (Amicon Ultra 0.5 mL, Millipore Corp., Burlington, MA, USA) and centrifuged at 12,000 rpm for 30 min to remove the denaturing reagents. The denatured proteins trapped on the filter were washed with 0.1 M Tris-HCl buffer (pH 8.5, 400 μL), followed by centrifugation at 12,000 rpm for 40 min. Further, 0.1 M Tris-HCl buffer (pH 8.5, 200 μL), 0.1 μg/μL trypsin (20 μL, Wako Pure Chemical Industries, Ltd., Osaka, Japan), and 0.1 μg/μL lysyl endopeptidase (20 μL, Wako Pure Chemical Industries, Ltd., Osaka, Japan) were added to the ultrafiltration tube and the denatured proteins were digested for 16 h at 37 °C on the filter. After digestion, the solution was centrifuged for 30 min at 12,000 rpm. The filtrate, which contained digested peptides (including glycopeptides), was transferred to a 10 kDa ultrafiltration tube (Amicon Ultra 0.5 mL, Millipore Corp., MA, USA) and centrifuged for 10 min at 12,000 rpm. Most glycopeptides were trapped on the filter membrane, whereas most non-glycosylated peptides passed through the filter [14]. The trapped glycopeptide fraction was washed with 10 mM ammonium acetate in 10% acetonitrile and 90% water (400 μL) and transferred into a 1.5 mL tube, followed by drying up by vacuum centrifugation. The glycopeptides trapped on the filter were recovered and analyzed by liquid chromatography and mass spectrometry (LC-MS).

### 4.4. Liquid Chromatography and Mass Spectrometry

The LC-MS data were acquired on a liquid chromatography system (Agilent HP1200, Agilent Technologies, Palo Alto, CA, USA) equipped with a C18 column (Inertsil ODS-4, 2 μm, 100 Å, 100 mm × 1.5 mm ID, GL Science, Tokyo, Japan) coupled with an electrospray ionization quadrupole time-of-flight (Q-TOF) mass spectrometer (Agilent 6520, Agilent Technologies, Palo Alto, CA, USA) [35]. Solvent A was 0.1% formic acid in water, and solvent B was 0.1% formic acid in 9.9% water and 90% acetonitrile. The glycopeptides were eluted at a flow rate of 0.15 mL/min at 40 °C with a following gradient program: 0 to 7 min, 15% to 30% solvent B; 7 to 12 min, 30% to 50% solvent B; and 2 min hold at 100% solvent B. The mass spectrometer was operated in negative mode with the capillary voltage of 4000 V. The nebulizing gas pressure was 30 psi and the dry gas flow rate was 8 L/min at 350 °C. The injection volume was 5 μL. LC-MS raw data were converted to CSV-type data using Mass Hunter Export (Agilent Technologies, Palo Alto, CA, USA) and peak positions (retention times and m/z) and peak intensities (areas) were calculated using R (R 3.2.2, R Foundation, Vienna, Austria). The errors of retention time and m/z were corrected by the internal standard (fetal calf fetuin) peaks and all peak intensities were aligned with the tolerance of 0.5 min (retention time) and 0.08 Da (m/z) by Maeker Analysis developed by LSI Medience Corp. [14].

### 4.5. Generation of 2D Barcodes

In total, 1712 glycopeptides were chosen from more than 30,000 detected peaks by following three steps: (1) removing low reproducibility peaks (CV (coefficient of variation) > 50%), (2) removing low reliability peaks (S/N (signal to noise) < 5), (3) removing isotope, adduct, and fragment ions. Subsequently, residual 1712 glycopeptide peaks were used for CSGSA diagnostics. QC samples were previously prepared by pooling several EOC and non-EOC samples. The expression values of 1712 glycopeptides were obtained by calculating the ratio between sample and QC values. The glycopeptide expression values were allocated to 41 × 42 cells of Excel sheet matrix (Microsoft, redmont, WA, USA) according to the order of elution time (Rt-based) or the order of PCA loadings (PCA-based). To generate Rt-based 2D barcodes, glycopeptide expression values were first aligned in the order of the elution time. Then, the values were allocated into 41 × 42 matrix of Excel sheet from top-left to bottom-right corner. To produce PCA-based 2D barcodes, PCA analysis (SIMCA 13.0.3, Umetrics, https://umetrics.com/products/simca) was first performed for 1712 glycopeptides and 351 examinees, and loading values of first and second components were obtained. Unit valiance scaling and zero average centering were performed for 1712 glycopeptides before PCA calculation. Then, the glycopeptide expression values were sorted based on the first loading values in the ascending order and divided into 41 groups with 42 glycopeptides. The glycopeptide expression values were further sorted in different groups based on the second loading values. The coordinates in 2D barcodes were determined in such a way that X coordinate represented the number of groups and Y represented the order in each group. After allocating 1712 glycopeptide expression values into Excel cells, the cell’s brightness was determined according to the criterion described in Appendix A. The generated 2D barcodes were converted to PNG format to let CNN learn.

### 4.6. Training CNN to Distinguish Between EOC and Non-EOC Patterns

MATLABand its deep learning toolbox (R2018b, Mathworks, Natick, MA, USA), which provides an environment of deep learning and neural network implementation, was used for training and testing of the CNN to discriminate early-stage EOC. AlexNet was obtained from Mathworks addin programs. We removed last 3 layers of AlexNet and added a new fully connected layer with two neurons, followed by softmax and classification layer, where we classified the data set into early-stage EOC or non-EOC. In this study, batch size was set to 5, max epochs were set to 30, and initial leaning rate was set to 0.0001. The training was performed on a desktop having NVIDIA GeForce GTX 960M DDR5 4096 MB GPU, which took around 10 min. The parameters of CNN were optimized using 60% of samples, and its validity was tested using 40% samples.

## 5. Conclusions

In this study, we showed that the pretrained CNN model AlexNet can recognize early-stage EOC using PCA-based 2D barcodes. The ROC further increased when 2D barcodes were generated based on the serum levels of CA125 and HE4. We observed that principal component analysis-based alignment of glycopeptides to generate 2D barcodes significantly increased the diagnostic accuracy of the method, and it reached 95.1% of ROC-AUC. We believe that this simple and low-cost method will reduce economical and physical burden on the patients and contribute to the improvement of the survival rates of EOC.

## Figures and Tables

**Figure 1 cancers-12-02373-f001:**
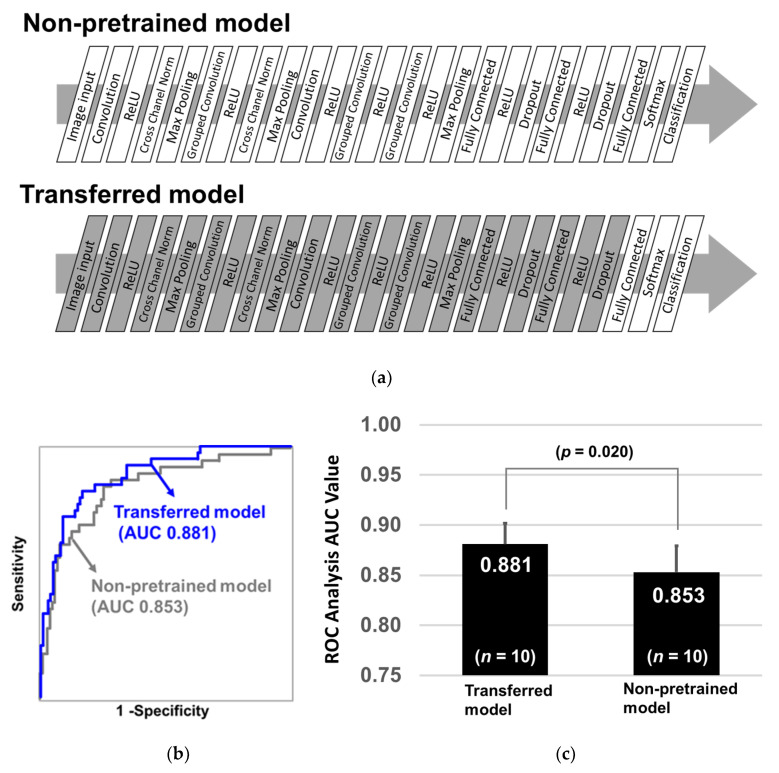
Diagnostic performance of non-pretrained and transferred AlexNet models. (**a**) Diagnostic performance (AUC (area under curve) of ROC (receiver operating characteristic)) of the non-pretrained and transferred AlexNet convolutional neural network (CNN) models was compared. The non-pretrained model was established by an original AlexNet framework with 25 fresh layers whose parameters were not optimized (Non-pretrained model). The composition of the transferred AlexNet model was the same as that of the non-pretrained model, but the parameters were already trained with the help of more than one million images from the ImageNet database; further, only last three layers were initialized (Transferred model). (**b**) White layer: non-trained, Gray layer: pretrained. After the two models were trained with 60% of the 351 samples, they were evaluated by the remaining 40% samples. ROC curves were shown with blue line (transferred) and gray line (non-pretrained). (**c**) This process was repeated 10 times and the *p*-values of the Wilcoxon rank sum test were calculated between two groups.

**Figure 2 cancers-12-02373-f002:**
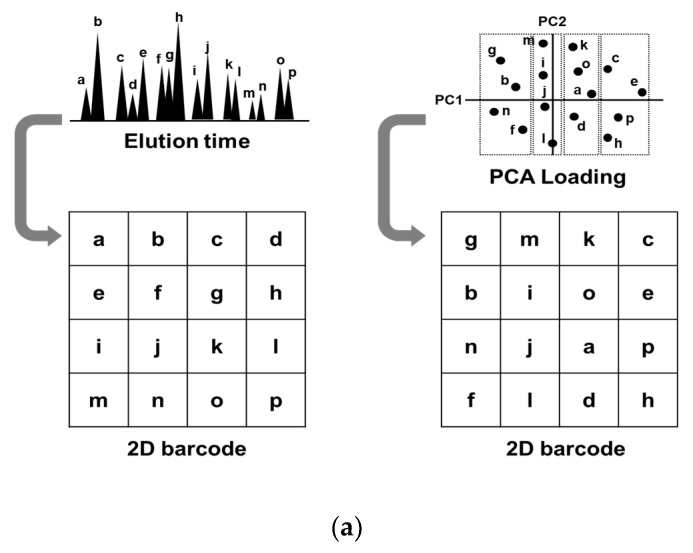
Alignment of glycopeptides to generate 2D barcode. (**a**) The Rt-based barcode was generated by aligning the glycopeptides from left-top to right-down by the order of elution time. The PCA-based barcode was developed by aligning the glycopeptides by the order of PCA loading values. (**b**) Typical Rt-based and PCA-based 2D-barcode images of the samples from patients with epithelial ovarian cancer (EOC) and of those with non-EOC. (**c**) After the two models were trained with 60% of the 351 barcodes, they were evaluated by the remaining 40% samples. The ROC curves are shown with a blue line (PCA-based) or with a gray line (Rt-based). (**d**) This process was repeated 10 times and the *p*-values of the Wilcoxon rank sum test were calculated between the two groups.

**Figure 3 cancers-12-02373-f003:**
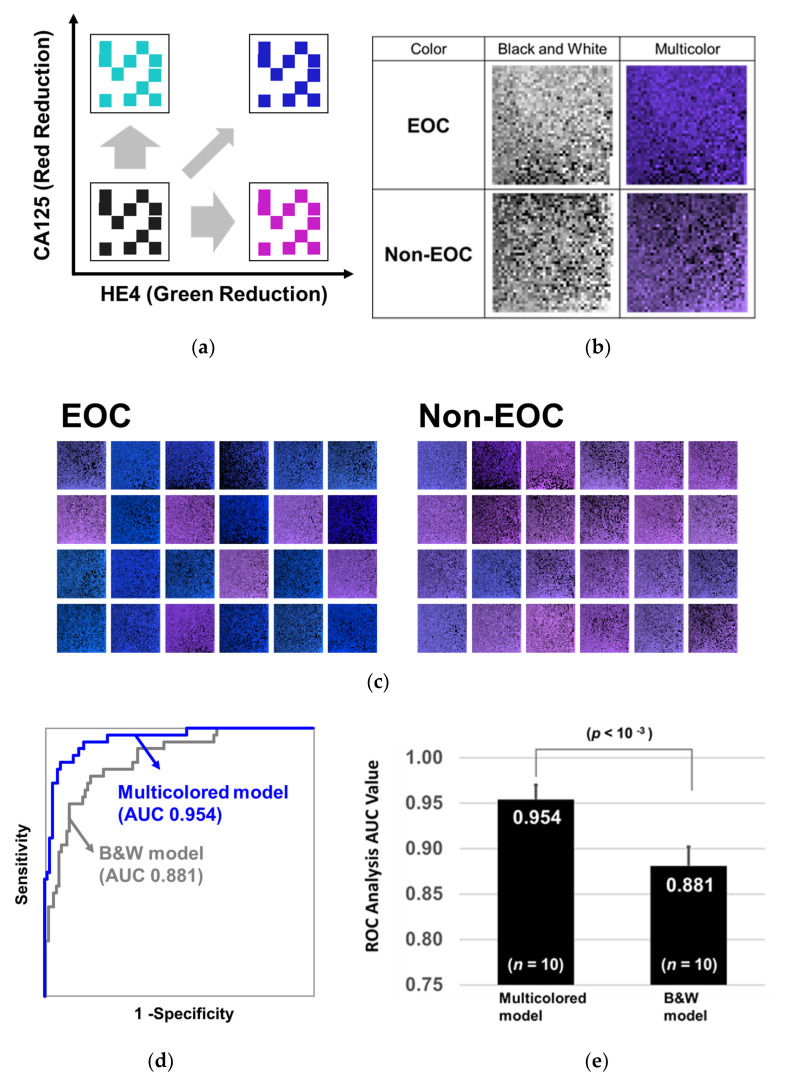
Coloring barcodes on the basis of serum levels of CA125 and HE4. (**a**) The non-multicolored barcodes were colored on the basis of the serum levels of CA125 and HE4. When the level of CA125 increases, the level of red in the barcode decreases, and when the level of HE4 increases, the level of green decreases. (**b**) Comparison of patients with EOC and those with non-EOC before and after the coloring of 2D-barcode images. (**c**) Typical 2D-barcode images of patients with EOC (left) and of those with non-EOC (right). (**d**) After the two models were trained with 60% of the 351 barcodes, they were evaluated by the residual 40% samples. The ROC curves are shown with a blue line (Multicolor) or with a gray line (Black and white). (**e**) This process was repeated 10 times and the *p*-values of the Wilcoxon rank sum test were calculated between the two groups.

**Figure 4 cancers-12-02373-f004:**
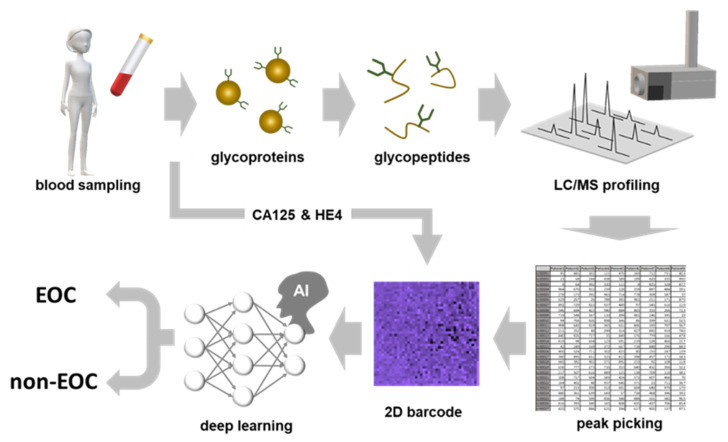
The whole scheme of artificial intelligence-based comprehensive serum glycopeptide spectra analysis (CSGSA-AI). The glycopeptides obtained by trypsin digestion of serum proteins were enriched and analyzed by the QTOF-type mass spectrometer. The expression levels of target glycopeptides were converted into 2D barcodes, and they were colored on the basis of serum CA125 and HE4 levels. After the CNN was trained by the supervised cases, it distinguished the barcodes generated from samples being screened into EOC or non-EOC.

**Table 1 cancers-12-02373-t001:** AlexNet framework.

Number	Layer Name	Role	Description
1	data	Image Input	227 × 227 × 3 images with “zero centering” normalization
2	conv1	Convolution	96 × 11, 11 × 3 convolutions with stride [4 4] and padding [0 0 0 0]
3	relu1	ReLU	ReLU
4	norm1	Cross Channel Normalization	cross channel normalization with 5 channels per element
5	pool1	Max Pooling	3 × 3 max pooling with stride [2 2] and padding [0 0 0 0]
6	conv2	Grouped Convolution	2 groups of 128 (5 × 5 × 48) convolutions with stride [1 1] and padding [2 2 2 2]
7	relu2′	ReLU	ReLU
8	norm2	Cross Channel Normalization	cross channel normalization with 5 channels per element
9	pool2	Max Pooling	3 × 3 max pooling with stride [2 2] and padding [0 0 0 0]
10	conv3	Convolution	384 3 × 3 × 256 convolutions with stride [1 1] and padding [1 1 1 1]
11	relu3	ReLU	ReLU
12	conv4	Grouped Convolution	2 groups of 192 3 × 3 × 192 convolutions with stride [1 1] and padding [1 1 1 1]
13	relu4	ReLU	ReLU
14	conv5	Grouped Convolution	2 groups of 128 3 × 3 × 192 convolutions with stride [1 1] and padding [1 1 1 1]
15	relu5	ReLU	ReLU
16	pool5	Max Pooling	3 × 3 max pooling with stride [2 2] and padding [0 0 0 0]
17	fc6	Fully Connected	4096 fully connected layer
18	relu6	ReLU	ReLU
19	drop6	Dropout	50% dropout
20	fc7	Fully Connected	4096 fully connected layer
21	relu7	ReLU	ReLU
22	drop7	Dropout	50% dropout
23	fc8	Fully Connected	1000 fully connected layer
24	prob	Softmax	softmax
25	output	Classification Output	crossentropyex with “EOC” and “Non-EOC”

Rectified Linear Unit.

**Table 2 cancers-12-02373-t002:** Sensitivity and specificity of CSGSA-AI.

CSGSA-AI (Test) (Cut Off = 0.5)						
		Condition		Total	PPV and NPV	
		EOC Stage1	Non			
CSGSA-AI	Pos	31	4	35	PPV	89%
	Neg	8	98	106	NPV	92%
Total		39	102	141		
Sensitivity and Specificity		Sens	Spec		Accuracy	
		79%	96%		91%	

**Table 3 cancers-12-02373-t003:** Study design.

Class		Cases	Age
EOC Stage I (*n* = 97)	Clear cell carcinoma	41	54.4 ± 12.9
	Mucinous carcinoma	14	
	Endometrioid carcinoma	28	
	Serous adenocarcinoma	13	
	Unclassified	1	
Non-EOC (*n* = 254)	Healthy	220	53.8 ± 11.4
	Uterine fibroid	20	
	Ovarian cyst	14	

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
