# Peer review of "Comprehensive Serum Glycopeptide Spectra Analysis Combined with Artificial Intelligence (CSGSA-AI) to Diagnose Early-Stage Ovarian Cancer"

_cancers, 2020, doi:10.3390/cancers12092373_

Round 1

Reviewer 1 Report

The authors submitted a paper that might be of interest due to early diagnosis of ovarian cancer, glycopeptide of serum, and novel AI technology.

It was thought that the poor prognosis of ovarian cancer could be sufficiently improved by improving the quality of early diagnosis, but the conventional measurement of CA125 did not exceed CA125.

The authors' long research on glycopeptide has provided a clue to address these limitations.

We expect a new large-scale trial for early diagnosis of ovarian cancer simply and inexpensively with CNN model AlexNET using PCA-based 2D barcodes

Author Response

We thank you for your comment and the opportunity to resubmit a revised copy of this manuscript. We would also like to take this opportunity to express our thanks to you for the positive feedback and helpful comments for correction or modification.

I understand that there are no requests to revise from you. I hope the other two reviewers accept our manuscript. 

Reviewer 2 Report

IIn this paper, the authors propose an artificial intelligence (AI)-based comprehensive serum glycopeptide spectra analysis (CSGSA-AI) method, in combination with convolutional neural network (CNN), to detect aberrant glycans serum samples of patients with EOC.

The work is very interesting, scientifically sound, and well structured. Its major flaw probably concerns the number of samples used (351, of which 211 for training): a very small number for a CNN, which usually needs thousands of samples to properly generalize the model. However, the results in terms of accuracy seems valid.

Overall, the quality of the writing is good, but there are numerous typographical errors in the text, including (not exhaustively):

- ln 111 pertaining -> pretraining
- ln 124 Student t test -> Student's t-test (in line 127 it is correct)
- ln 140 remining -> remaining

Hence, please perform a thorough grammar check. Moreover, the authors repeats too many times that the samples are subdivided in 60% train and 40% test: these parts should be better harmonised.

In addition, bibliography needs to be improved:

- ln 128: this validation approach is usually called "repeated hold-out", or "monte carlo cross-validation": please mention it and indicate a bibliographical reference;

- ln 241: which previous study? please add a reference;

- In the Introduction, some references to recent work which exploit CNN approaches in other topics should also be included, for example:

 1) pedestrian recognition: Occlusion-aware R-CNN: Detecting Pedestrians in a Crowd;
 2) surveillance: Efficient CNN based summarization of surveillance videos for resource-constrained devices;
 3) market forecasting: Deep learning and time series-to-image encoding for financial forecasting

Finally, the Conclusions should be expanded, as they are too concise. However, I believe that, once these points have been addressed, the manuscript is worthy of publication in this prestigious journal.

Author Response

  We thank you for your comment and the opportunity to resubmit a revised copy of this manuscript. We would also like to take this opportunity to express our thanks to you for the positive feedback and helpful comments for correction or modification.

We show the answers for the advice we were given from you.

(Request 1) There are numerous typographical errors in the text, including (not exhaustively):

- ln 111 pertaining -> pretraining    

- ln 124 Student t test -> Student's t-test (in line 127 it is correct)

- ln 140 remining -> remaining

(Answer 1)

 I corrected all the misspellings. Thank you.

(Request 2)

Moreover, the authors repeat too many times that the samples are subdivided in 60% train and 40% test: these parts should be better harmonized.

(Answer 2)

 I replaced these explanations with “the diagnostic performance was evaluated with the repeated holdout validation method”.

(Request 3)

In addition, bibliography needs to be improved:

- ln 128: this validation approach is usually called "repeated hold-out", or "monte carlo cross-validation": please mention it and indicate a bibliographical reference.

(Answer 3)

 To clarify this process, I added the next explanation. “To evaluate the diagnostic performance of the two models, we calculated ROC-AUCs using the repeated holdout validation method as described next.”

(Request 4)

- ln 241: which previous study? please add a reference.

(Answer 4) 

 I added the reference to the “previous study”.

(Request 5)

- In the Introduction, some references to recent work which exploit CNN approaches in other topics should also be included, for example:

 1) pedestrian recognition: Occlusion-aware R-CNN: Detecting Pedestrians in a Crowd;

 2) surveillance: Efficient CNN based summarization of surveillance videos for resource-constrained devices;

 3) market forecasting: Deep learning and time series-to-image encoding for financial forecasting

(Answer 5)  

I added the references of the past contributions of CNN as below.

“Recently, artificial intelligence (AI), in particular image recognition using convolutional neural network (CNN), has emerged to distinguish objects in images [19], and now contributes on pedestrian recognition[20], surveillance videos[21] or financial forecasting[22].”

(Request 6)

Finally, the conclusions should be expanded, as they are too concise.

(Answer 6)  

We expand conclusions as below, however, since Cancers has both “Discussion” and “Conclusions” parts, and we mainly discuss our future prospects in discussion parts, we stated a brief conclusion in Conclusion part. I am happy if you could understand it.

 “ In this study, we showed that the pretrained CNN model AlexNet can recognize early-stage EOC using PCA-based 2D barcodes. The ROC further increased when 2D barcodes were generated based on the serum levels of CA125 and HE4. We observed that principal component analysis-based alignment of glycopeptides to generate 2D barcodes significantly increased the diagnostic accuracy of the method, and it reached 95.1 % of ROC-AUC. We believe that this simple and low-cost method will reduce economical and physical burden on the patients and contributes to the improvement of the survival rates of EOC.”

Reviewer 3 Report

The manuscript “Comprehensive Serum Glycopeptide Spectra 2 Analysis Combined with Artificial Intelligence 3 (CSGSA-AI) to Diagnose Early-Stage Ovarian Cancer” strives to develop an artificial intelligence-based comprehensive serum glycopeptide spectra analysis method in combination with convolutional neural network to detect aberrant glycans in serum samples of patients with EOC. The technique appears to have been developed carefully and may have value. But there are several concerns about the presentation of the report.

The presence of aberrant serum sugar chains in patient sera is not per se surprising. However, the consistency of such deviations is unclear. Would the same patient, measured at various times of day, after diverse activities, or in distinct states of fasting, show the same pattern of deviation from healthy controls? How cancer- or patient-specific are the changes? If there is literature evidence, it should be discussed in some detail. Otherwise, a set of experiments would be helpful as such information will be important for evaluating the barcode.

The writing is unclear in multiple places.
1) The calculation of significance in the evaluation of ROC curves does not usually employ Student’s t-test. The authors need to justify their choice, preferably employ one of the more commonly used tests. If the t-test pertains to the areas under the curve from runs with various training and test sets, the authors need to state that clearly and confirm that the data were normally distributed and had equal variance.
2) The development of the 2D barcode is unclear. One dimension is the liquid chromatography elution time, the other dimension comes from the principal component analysis. The latter was done in 2 dimensions and first and second loading scores were obtained. There needs to be more clarity how the principal component analysis was converted into the second dimension of the barcode.
3) The parameters used in the principal component analysis are not expressedly stated. It is also important to ascertain that there is no overlap with the characteristics that determine the liquid chromatography elution time.

Minor points:
The number of acronyms is unreasonably excessive. The manuscript is difficult to read.
A 41 x 42 matrix has 1722 entries. With the expression values of only 1712 glycopeptides, there are empty slots. Consistently, for the black-and-white depictions in Figure 3 that seems to be the case on the bottom right corner. The colored matrices are filled, which is misleading.

Author Response

We  thank you for your valuable comments and the opportunity to resubmit a

revised copy of this manuscript. We would also like to take this opportunity to express our thanks to you for the positive feedback and helpful comments for correction or modification.

We believe have resulted in an improved revised manuscript, which you will find uploaded alongside this document. The manuscript has been revised to address your comments. We very much hope the revised manuscript is accepted for publication in Cancers.

(Request 1)

The presence of aberrant serum sugar chains in patient sera is not per se surprising. However, the consistency of such deviations is unclear. Would the same patient, measured at various times of day, after diverse activities, or in distinct states of fasting, show the same pattern of deviation from healthy controls? How cancer- or patient-specific are the changes? If there is literature evidence, it should be discussed in some detail. Otherwise, a set of experiments would be helpful as such information will be important for evaluating the barcode.

(Answer 1)

Current cancer markers with sugar chains, such as CA19-9, AFP-L3, SLex, and CA125 are relatively stable and are not influenced by circadian rhythm, diet, or activities. However, as you indicated, it is known that the levels of some glycoproteins are changed by several factors, therefore we are thinking that the validation is necessary. I added this issue in discussion part as below.

“Some issues we should address remain before we realize this technology as a clinical test. It is known that the levels of some glycoproteins in serum are influenced by diet, sex, or circadian activities, therefore the stability of the target glycopeptides should be validated.”

(Request 2)

The calculation of significance in the evaluation of ROC curves does not usually employ Student’s t-test. The authors need to justify their choice, preferably employ one of the more commonly used tests. If the t-test pertains to the areas under the curve from runs with various training and test sets, the authors need to state that clearly and confirm that the data were normally distributed and had equal variance.

(Answer 2)

 I recalculated p-values with the non-parametric evaluation method, “Wilcoxon rank-sum test”, and I replaced these values with new values calculated by Wilcoxon rank-sum test.

(Request 3)

The development of the 2D barcode is unclear. One dimension is the liquid chromatography elution time, the other dimension comes from the principal component analysis. The latter was done in 2 dimensions and first and second loading scores were obtained. There needs to be more clarity how the principal component analysis was converted into the second dimension of the barcode.

(Answer 3)

The process of developing 2D barcode is somewhat complicated. So, I improved Figure 2a to give readers an intuitive understanding.

Rt-based and PCA-based 2D barcodes are completely independent and different, and Rt-based 2D barcode does not use PCA information.

To generate Rt-based 2D-barcodes, glycopeptide expression values were first aligned in the order of the elution time. Then, the values were allocated into 41 × 42 matrix of Excel sheet from top-left to bottom-right corner as shown in Figure 2a. On the other hand, to produce PCA-based 2D-barcodes, the glycopeptide expression values were sorted based on the first loading values in the ascending order and divided into 41 groups with 42 glycopeptides. The glycopeptide expression values were further sorted in different groups based on the second loading values. The coordinates in 2D barcodes were determined in such a way that X coordinate represented the number of groups and Y represented the order in each group.

(Request 4)

The parameters used in the principal component analysis are not expressedly stated. It is also important to ascertain that there is no overlap with the characteristics that determine the liquid chromatography elution time.

(Answer 4)

I added the following sentence to avoid reader’s misunderstandings. “Unit valiance scaling and zero average centering were performed for 1712 glycopeptides before PCA calculation.”

(Request 5)

The number of acronyms is unreasonably excessive. The manuscript is difficult to read.

(Answer 5)

I spelled out some acronyms as below.

PPV: Positive predictive value

NPV: Negative predictive value

GANs: Generative adversarial networks

However, we keep using these acronyms with following reasons.

CSGSA: We have already published “CSGSA” in Cancers in 2019

CA125: This acronym is widely used.

HE4: This acronym is widely used.

CNN: This acronym is widely used.

EOC: This word was used 70 times in the manuscript.

ROC: This acronym is widely used.

AUC: This acronym is widely used.

AI: This acronym is widely used.

PCA: This acronym is widely used.

OPLS-DA: the acronym form “OPLS-DA” is much popular than the spelled-out form “orthogonal partial least squares discriminant analysis”.

(Request 5)

A 41 x 42 matrix has 1722 entries. With the expression values of only 1712 glycopeptides, there are empty slots. Consistently, for the black-and-white depictions in Figure 3 that seems to be the case on the bottom right corner. The colored matrices are filled, which is misleading.

(Answer 5)

I added following statement.  “Since a 41 x 42 matrix has 1722 cells, and 10 blanks are generated on the bottom right corner with the expression values of 1712 glycopeptides, the blank cells were replaced with white.”

 “Ten blanks generated on the bottom right corner were filled with the colors defined by both CA125 and HE4.”

Round 2

Reviewer 3 Report

The authors have made an effort to address the concerns raised in the initial review. I support the acceptance of the manuscript.